# Development of Pollen Parent Cultivar-Specific SCAR Markers and a Multiplex SCAR-PCR System for Discrimination between Pollen Parent and Seed Parent in Citrus

**DOI:** 10.3390/plants12233988

**Published:** 2023-11-27

**Authors:** Sang Suk Kim, Seung Gab Han, Yo Sup Park, Suk Man Park, Cheol Woo Choi, Su Hyun Yun, Dong Hoon Lee, Seong Beom Jin

**Affiliations:** Citrus Research Institute, National Institute of Horticultural & Herbal Science, RDA, Jeju 63613, Republic of Korea; sskim0626@korea.kr (S.S.K.); skhan@korea.kr (S.G.H.); yspark1219@korea.kr (Y.S.P.); babau2000@korea.kr (S.M.P.); cwchoi7@korea.kr (C.W.C.); yunsh04@korea.kr (S.H.Y.); chocho90@korea.kr (D.H.L.)

**Keywords:** Asumi, Asuki, Korea, crossbreeding, morphology, hybrid, seedling

## Abstract

This study discusses the challenge of distinguishing between two high-quality mandarin cultivars, ‘Asumi’ and ‘Asuki’, which have been introduced and cultivated in Korea after being developed through crossbreeding in Japan. Owing to genetic similarities resulting from crossbreeding between the same parent cultivars, it is challenging to differentiate them morphologically at the seedling stage. This difficulty poses challenges for cultivation and harvesting on farms. To address this issue, we developed a method using sequence characteristic amplification region (SCAR) markers for rapid and accurate differentiation between the two cultivars. We selected specific primer sets from random amplified polymorphic DNA–SCAR combinations and sequence-related amplified polymorphism contrast markers. The multiplex PCR system using these molecular markers was able to identify 16 mandarin cultivars, including ‘Asumi’ and ‘Asuki’, among 30 cultivars. The use of these SCAR markers is expected to enhance citrus cultivation by accurately identifying mixed cultivars and facilitating proper harvest timing for citrus distribution. Additionally, the markers can help identify the genetic traits of hybrid varieties at the seedling stage.

## 1. Introduction

The genus *Citrus* belongs to the subfamily Aurantioideae of the family Rutaceae [1], and is one of the most important crops cultivated worldwide [2,3]. Generally, citrus cultivars are classified into five citrus groups for commercial purposes, including mandarins (*C. reticulata* and *C. unshiu* Marc.), sweet oranges (*C. sinensis* [L.] Osb.), lemons (*C. limon* Burm. f.), limes (*C. aurantifolia* L.), and grapefruits (*C. paradisi* Macf.) [4,5].

Mandarins are varieties with relatively small, round fruits, thin peels, and minimal white fiber inside, making them easy to peel, with easily dividable segments [6,7]. Owing to these key characteristics, consumer preferences have rapidly shifted toward delicious mandarin varieties that are seedless and easy to peel [6,7,8,9,10,11,12,13,14,15,16]. Therefore, mandarin varieties are commonly used as either pollen parents or seed parents in cultivar breeding through hybridization [17,18,19].

In this context, the high-quality citrus cultivars ‘Asumi’ and ‘Asuki’, developed by crossbreeding the seedless, male sterile cultivars ‘Okitsu 46-gu’ and ‘Harumi’ at the National Institute of Fruit Tree Science in Japan, have been introduced and cultivated by domestic farmers in South Korea. However, distinguishing between these two cultivars morphologically at the seedling stage is difficult [20,21,22,23]. They can only be primarily distinguished based on differences in their key fruit characteristics at the mature stage. Although ‘Asumi’ ripens from late January to mid-February, and ‘Asuki’ ripens from March onward, both cultivars share similar characteristics, such as a high sugar content of approximately 15–16°Bx and low acidity levels of approximately 1% [24,25]. ‘Asumi’, an excellent Japanese citrus cultivar, was introduced to Korea in around 2015 through a domestic nursery sales company. Presently, it is cultivated by approximately 82 farms on 27 hectares, although this is unofficial data [26]. However, due to patent royalty issues with Japan, its distribution was initially hindered. In 2020, SPfresh, a Korean fruit distribution and sales company, secured a sales permit contract with the Japan National Agriculture and Food Research Organization, thereby allowing for the official distribution and sale of ‘Asumi’ (https://m.segye.com/view/20201111508897 (accessed on 11 November 2020)). In the Korean market, ‘Asumi’ is marketed under the name “Surahyang,” whereas ‘Asuki’ is sold as “Sinbihyang.”

Complaints have emerged regarding the production of fruit with distinct characteristics from seemingly identical trees, suggesting the inadvertent introduction of the ‘Asuki’ cultivar during the process of introducing ‘Asumi’ to Korea. Owing to genetic recombination during crossbreeding between the same parent cultivars, morphological differentiation between these two cultivars is challenging [24,25,27,28]. Consequently, the development of molecular markers that can easily and accurately identify the cultivars is imperative to prevent farmers from producing low-quality citrus due to the inability to determine the optimal harvest time [22,27,29,30,31].

Various molecular markers have been used for distinguishing citrus cultivars, such as random amplified polymorphic DNA (RAPD), restriction-fragment-length polymorphisms, amplified-fragment-length polymorphisms, expressed sequence tag–simple sequence repeats (SSRs), genomic SSRs, cleaved amplified polymorphic site (CAPS) markers [22,30,32,33,34,35,36,37], single-nucleotide polymorphisms (SNPs) [38], and sequence-related amplified polymorphism–sequence characteristic amplification region (SRAP-SCAR) markers [39]. Insertion–deletion (InDel) markers, developed based on the whole-genome resequencing of the Satsuma mandarin, have also emerged in recent years [9]. CAPS, SNP, and InDel markers are widely used for distinguishing morphologically similar cultivars and are relatively straightforward to analyze [29,30,40]. However, the development cost of these markers is high, and they necessitate restriction enzyme digestion [27]. Conversely, RAPD markers have issues with reproducibility [41]. However, SCAR markers based on RAPD markers offer distinct advantages, including high stability, reproducibility, and operational simplicity [39,41,42,43,44,45]. Additionally, the multiplex molecular-marker-assisted PCR system method [46] using SCAR markers, including control markers, can provide a more accurate and time- and cost-saving approach for identifying varieties at the seedling stage [38].

Herein, we aimed to develop a rapid and precise SCAR marker to address the challenge of identifying citrus varieties that are difficult to distinguish in cultivation, harvest, and quality management owing to their uncertain classification. We also aimed to use mandarin variety-specific multiple SCAR markers to identify which of the parental genetic traits was most influenced by genetics. By analyzing the nucleotide sequence of the UBC218 product, which exhibits specific amplification in the ‘Asumi’ cultivar, SCAR primers were designed. Multiple primers were investigated to identify a rapid and reliable marker capable of accurately distinguishing between the mixed ‘Asumi’ and ‘Asuki’ cultivars within a short timeframe. Furthermore, we investigated whether the selected SCAR markers were specifically amplified in any citrus cultivars.

## 2. Results

### 2.1. Fruit Quality Characteristics

The quality characteristics of the fruit of ‘Asumi’ (Figure 1a) and ‘Asuki’ (Figure 1b) were compared and analyzed in this study. ‘Asumi’ was introduced and cultivated in Korea following the crossbreeding of ‘Okisu No. 46′ (*C.* hybrid ‘Okisu No. 46′) and ‘Harumi’ (*C.* hybrid ‘Harumi’) in Japan, and ‘Asuki’ was another variety with a similar origin. Morphologically, it is challenging to differentiate between these two varieties as they were developed through crossbreeding the same parent varieties. A comparative analysis of the sugar content and acidity level between ‘Asumi’ and ‘Asuki’ revealed that the sugar contents of ‘Asumi’ (15.58 Brix) and ‘Asuki’ (15.14 Brix) were similar, although the former had a lower acidity level than the latter (1.1% vs. 1.92%) (Table 1).

### 2.2. Selection of Specific SCAR Markers and Control Markers for ‘Asumi’

To distinguish between ‘Asumi’ and ‘Asuki’, specific SCAR markers and control markers were developed. The RAPD UBC 218 primer yielded an amplified product specific to ‘Asumi’ (Figure 2). Through cloning and sequencing analysis, we confirmed that this amplified product consisted of 464 base pairs (Appendix A). By comparing this specific sequence with registered gene sequences in the National Center for Biotechnology Information (NCBI) database (www.ncbi.nlm.nih.gov, accessed on 30 October 2023) using the BLAST program, we found that its highest similarity (67%) was with the *Citrus clementina* putative uncharacterized protein DDB_G0290521 (GenBank Acc. No. XM 024187014) from the clementina cultivar. Based on this analysis, we designed seven sets of single SCAR primers, each generating PCR products of approximately 200–457 bp (Appendix A). PCR using these primers led to the identification of a specific SCAR primer combination (primer 16) that produced an amplified product in ‘Asumi’ but not in ‘Asuki’ (Figure 3 and Appendix A).

To ensure an equal DNA content between ‘Asumi’ and ‘Asuki’, we used the SRAPF3/R5 primer for amplification, which resulted in a product obtained from both varieties (Figure 4). The cloning and sequencing analysis of this product revealed a sequence consisting of 751 base pairs (Appendix A). The comparison of this sequence with registered gene sequences in the NCBI database using BLAST revealed that its highest similarity (96%) was with an uncharacterized gene [*Citrus sinensis* uncharacterized LOC107177919 (LOC107177919)] of the *Citrus sinensis* cultivar. Based on this analysis, we designed five sets of single SRAP-SCAR primer combinations, each generating PCR products of approximately 200–700 bp (Table 1; Materials and Methods). PCR using these primers led to the identification of a specific primer combination (primer 8) that produced an amplified product in both ‘Asumi’ and ‘Asuki’ (Figure 5 and Appendix A).

### 2.3. Multiplex SCAR Marker Development

Multiplex PCR was performed using selected primers based on the RAPD-SCAR markers specific to ‘Asumi’, which were identified from 16 combinations (Appendix A). Primers No. 1 and No. 3, No. 1 and No. 7, No. 12 and No. 13, and No. 13 and No. 15 were mixed and used in a single PCR to amplify multiple products simultaneously. Figure 6 shows that, when primers No. 1 (457 bp) and No. 3 (approximately 381 bp) were used in combination, ‘Asumi’ produced two amplification products. Similarly, ‘Asumi’ produced amplification products when primers No. 1 (457 bp) and No. 7 (approximately 393 bp) were used together. Additionally, combining primers No. 12 (approximately 311 bp) and No. 13 (457 bp) resulted in the simultaneous amplification of both products. However, when primers No. 13 (457 bp) and No. 15 (approximately 370 bp) were used together, nonspecific amplification products were observed in addition to the expected product (Figure 6). Therefore, this primer combination was excluded from further experiments. Subsequently, multiplex PCR was performed using the three primer combinations (No. 1/No. 3 and No. 1/No. 7; No. 13/No. 12) specific to ‘Asumi’ and the two primer combinations (SRAP F2/R2 and F2/R3) that amplified products in all varieties, which were used as control markers. As shown in Figure 7, the negative control showed no amplification products, whereas both ‘Asumi’ and ‘Asuki’ produced products specific to the control markers. However, ‘Asumi’ produced three amplification products, whereas ‘Asuki’ yielded only one amplification product (Figure 7).

### 2.4. Application of Multiplex PCR and Commercialization in Citrus Farms 

Based on these findings, multiplex PCR was used to distinguish ‘Asumi’ from 24 trees collected from six farms cultivating both ‘Asumi’ and ‘Asuki’. Three specific RAPD-SCAR marker combinations for ‘Asumi’ (No. 1/No. 3, No. 1/No. 7, and No. 13/No. 12) and one SRAP-SCAR control marker combination (SRAP F2/R3) capable of amplifying both ‘Asumi’ and ‘Asuki’ were used. As shown in Figure 8, the second and third farms were confirmed to cultivate ‘Asumi’, whereas the fourth farm exclusively cultivated ‘Asuki’. The first, fifth, and sixth farms were found to cultivate a mixture of ‘Asumi’ and ‘Asuki’ (Figure 8).

### 2.5. Validation of Mandarin Cultivar-Specific SCAR Markers

To confirm that only cultivars derived from the same mandarin pollen parent genotype as ‘Asumi’ at the seedling stage, both mandarin cultivars and hybrid cultivars were used as seed parents or pollen parents (Appendix A). PCR analysis was performed using the six selected primer combinations from the RAPD-SCAR markers and control SRAP-SCAR markers (two combinations). ‘Asumi’ (S1) and 15 other cultivars (S8–S22) derived from the same mandarin pollen or seed parent produced the three expected products (Figure 9). In contrast, the remaining 14 cultivars (S1–S7 and S23–S30), including Satsuma mandarin and orange cultivars, such as ‘Asuki’, and crossbreeding cultivars not influenced by the same mandarin pollen parent genotype as ‘Asumi’, yielded only the control marker amplification products. Moreover, no amplification products were observed in the negative control (lane NC in Figure 9), indicating the absence of primer contamination.

Additionally, PCR was performed using ‘Asumi’ identification markers on 13 citrus cultivars commonly referred to as mandarins owing to their key characteristics (Appendix A). The results indicated that, except for three varieties (‘Kinokuni’, ‘Byungkyul’, and ‘Page’), the mandarin varieties exhibited the presence of three amplification products (Figure 10).

## 3. Discussion

Generally, mandarin varieties are used for breeding as pollen parents or seed parents for hybridization owing to their excellent main characteristics [5,6,7,8,9,10,11,12,13,14,15,16,17,18,19]. Mandarin cultivars, including Satsuma mandarin, were used for crossbreeding in this study. Although these cultivars share a name with general mandarin varieties, they differ in main characteristics as they are male sterile, seedless, and polyembryonic [18]. Hybrids bred through crossbreeding with these varieties are difficult to distinguish morphologically at the seedling stage between seed parent-derived nucellar seedlings and zygotic seedlings [18]. Therefore, this study first developed a simple identification technique that can easily distinguish the ‘Asumi’ and ‘Asuki’ varieties, which pose challenges due to incorporation at the introduction stage—a major concern for domestic farms. The selected markers were then used to identify pollen parents. Among the hybrid varieties used as seed parents, our aim was to prevent problems for citrus farmers in the cultivation and harvesting process by establishing, at the seedling stage, whether the varieties are derived from the Satsuma mandarin genotype or from the mandarin variety-specific genotype.

In this study, the sugar content and acidity level of ‘Asumi’ and ‘Asuki’ were compared by harvesting the fruits in early February. The results revealed that both varieties had a sugar content exceeding 15 Brix, with ‘Asumi’ exhibiting an acidity level of 1.1% and ‘Asuki’ displaying a higher acidity level of 1.92%. These findings align with those of previous studies [24,25] on the characteristics of these varieties. Hiroyuki et al. [24] reported that ‘Asuki’ has a ripening period from the end of February to the end of March, during which the sugar content is 14.7–15 Brix, and the acidity reaches 1.37%, although it decreases to <1% between March and April. However, in the present study, ‘Asuki’ harvested in early February exhibited a sugar content of >15 Brix and a high acidity level of 1.92%. Therefore, it can be inferred that ‘Asuki’ demonstrates adaptability depending on the cultivation area and conditions [24].

Given that ‘Asumi’ and ‘Asuki’ were derived from the same parental cultivars (‘Okitsu 46 gou’ × ‘Harumi’), distinguishing them morphologically before fruit ripening is challenging. Consequently, farms with the mixed cultivation of these two varieties face the risk of harvesting and distributing low-quality citrus fruits due to harvest timing errors. The early identification of these varieties using molecular markers could help mitigate the damage caused by variations in their growth periods. Previous studies [29,40,47,48] employed the comparative analysis of citrus genome sequences to identify multiple InDel polymorphisms, from which polymorphic InDel markers were developed using PCR. However, because the development and implementation of this method requires significant time and financial resources, a simple yet cost-saving identification technique is required.

In this study, the ‘Asumi’ variety was specifically amplified using the RAPD UBC 218 primer, and the resulting product was used for SCAR marker cloning. Seven combinations of SCAR primers were designed, with five used in multiplex PCR to identify four primer combinations capable of distinguishing between ‘Asumi’ and ‘Asuki’. Additionally, the SRAPF3/R5 primer combination, which amplified a product in both ‘Asumi’ and ‘Asuki’, was used to develop a control marker to ensure the presence of equal DNA amounts. Five combinations of SRAP-SCAR primers were designed, and the eighth combination, serving as control primers amplifying both ‘Asumi’ and ‘Asuki’, was selected. Furthermore, six primer combinations were evaluated for their suitability in performing multiplex PCR, incorporating four RAPD-SCAR markers and two combinations of SRAP-SCAR markers (SRAP-SCAR F2/R2 and F2/R3) specifically chosen for discriminating ‘Asumi’. The results demonstrated that ‘Asumi’ yielded three amplification products, whereas ‘Asuki’ produced one amplification product using three multiplex PCR primer combinations. One primer combination resulted in nonspecific amplification and was excluded from further analysis. Moreover, the discriminability of the selected markers was tested upon application to both ‘Asumi’ and ‘Asuki’ cultivation farms. Compared with the use of primer pairs for variety discrimination, the multiplex PCR method established in this study is expected to save time and reduce costs [38,49]. Although Das et al. [50] reported the successful development of a multiplex PCR technique using a combination of two selected SCAR markers to obtain two amplified products, there have been no reports of developing a multiplex PCR method combining three sets of variety identification markers and control markers.

Additionally, the selected ‘Asumi’ identification markers were used to identify cultivars from a total of 30 citrus species, including mandarins, oranges, and mandarin hybrids, which also included the common Satsuma mandarin species and 13 mandarin cultivars (Appendix A). Therefore, it was possible to simultaneously distinguish specific mandarin varieties used as seed parents or pollen parents, Satsuma mandarin species, and hybrids within their lineage. Generally, hybrid plants are presumed to exhibit differences in major characteristics [24,25] among identical parent hybrid varieties due to genetic changes during the recombination of pollen or seed parent genes [28,40]. Furthermore, the markers identified in this study can also be used to determine whether the major characteristic genes of the domestically bred hybrid variety ‘Minihyang’ (‘Kinokuni’ × ‘Ootaponkan’) [17,51] originated from pollen parents or seed parents. In light of the results of this study and those of a previous study [40], it is possible to establish whether a hybrid mandarin variety used as a pollen parent or seed parent belongs to the Satsuma mandarin lineage, originates from a specific mandarin lineage, is a true hybrid zygotic seedling, or has nucellar embryony originated from a seed parent lineage [18]. These results can further be used to investigate the phylogenetic origin of varieties more efficiently.

## 4. Materials and Methods

### 4.1. Analysis of Fruit Quality Characteristics

To assess fruit quality, the sugar content and acidity were compared between ‘Asumi’ and ‘Asuki’ using fruit harvested in early February. The sugar content of the fruits was measured using a digital sugar analyzer (PAL-1, ATAGO CO., LTD, Saitama, Japan). The acidity (% citric acid in fresh weight) of the fruits was determined by mixing 5 mL of flesh juice with 0.2 mL of 1% phenolphthalein and titrating it with 0.01 N NaOH solution to the endpoint. The soluble solids content and titratable acidity were measured using 20 replicates.

### 4.2. Plant Materials and Whole-Genome DNA Extraction

To develop SCAR markers for the early discrimination of the ‘Asumi’ (*Citrus* spp. ‘Asumi’) and ‘Asuki’ (*Citrus* spp. ‘Asuki’) varieties, developed by crossing the Japanese mandarin ‘Okitsu wase’ (*Citrus unshiu* ‘Okitsu wase’) and ‘Harumi’ (*Citrus* spp.) and subsequently introducing it to Korea, leaf tissues were collected from these varieties by SPfresh, a fruit distribution company in Korea, as samples for marker selection. Additionally, to test the feasibility of using the selected SCAR markers in agriculture, leaf tissues of ‘Asumi’ and ‘Asuki’ were collected from six domestic farms cultivating both varieties and stored at −20 °C for subsequent experiments.

Additionally, the citrus specimens used in this study included a total of 43 varieties, comprising 4 species of Satsuma mandarin cultivars (*C. unshiu* ‘Okitsu wase’, *C. unshiu* ‘Miyagawa wase’, *C. unshiu* ‘Nichinan 1gou’, and *C. unshiu* ‘Haraejosaeng’), 20 cultivars of *Citrus reticulata* mandarins (‘Wilking’, ‘Nowa’, ‘Lee’, ‘Encore’, ‘Ootaponkan’, and ‘Hayaka Ponkan’) as well as oranges (‘Washingon navel’) and mandarin hybrids (*C.* hybrid ‘Harumi’, *C.* hybrid ‘Kiyomi’, *C.* hybrid ‘Beni Madonna’, *C. reticulata* ‘Ootaponkan’, *C.* hybrid ‘Shiranuhi’, and the *C.* hybrid ‘Kanpei’) (Appendix A). These samples were used for the application of the selected SCAR markers. Whole-genome DNA was extracted from the samples using an Automatic Nuclear Extraction Kit (MX 16; Promega, Madison, WI, USA) and stored at −20 °C for future use.

### 4.3. PCR for Breed-Specific SCAR Markers and Control Markers

To select breed-specific markers and control markers for ‘Asumi’, RAPD and SRAP techniques were used. To identify ‘Asumi’-specific RAPD-SCAR markers, 200 University of British Columbia (UBC; Vancouver, BC, Canada) primers, each consisting of 10 nucleotides, were obtained from Bioneer Corp. (Daejeon, Korea), and PCR was performed. The PCR reaction mixture (20 μL) was prepared with AccuPower^®^ Multiplex PCR PreMix (Bioneer Corp.) containing 250 μM of dNTP, 1.5 mM of MgCl_2_, 1.0 unit Taq DNA polymerase, 10 mM of Tris-HCl (pH 9.0), and 40 mM of KCl. Each reaction included 20–50 ng of genomic DNA and 5 μM of primer. The PCR reaction was initiated by denaturing the DNA at 94 °C for 5 min, followed by 45 amplification cycles at 94 °C for 1 min, 38 °C for 2 min, and 72 °C for 1 min, with a final extension at 72 °C for 5 min.

To select control markers for comparative analysis, SRAP PCR was performed as described by Kang et al. [39]. The SRAP primer composition consisted of SRAP-F3 (5ʹ-TGAGTCCAAACCGGAAG-3′) and R5 (5ʹ-GACTGCGTACGAATTAGA-3ʹ). The PCR reaction mixture (20 μL) was prepared with AccuPower^®^ Multiplex PCR PreMix (Bioneer Corp.) containing 250 μM of dNTP, 1.5 mM of MgCl_2_, 1.0 unit Taq DNA polymerase, 10 mM of Tris-HCl (pH 9.0), and 40 mM of KCl. Each reaction included 20–50 ng of genomic DNA and 0.5 μM of primer. The PCR reaction involved an initial denaturation step at 94 °C for 5 min, followed by 5 amplification cycles at 94 °C for 1 min, 35 °C for 1 min, and 72 °C for 1 min, and then 35 amplification cycles at 94 °C for 1 min, 50 °C for 1 min, and 72 °C for 1 min, with a final extension at 72 °C for 10 min. These PCR products were subsequently analyzed using the QiAxcel advanced system (Qiagen, Hilden, Germany) electrophoresis device.

### 4.4. Cloning of PCR Amplification Products

To develop specific SCAR markers and comparison markers, PCR was performed using the UBC 218 primer set, specifically targeting ‘Asumi’ based on RAPD analysis, and the F3/R5 primer set selected for comparison markers through the SRAP analysis. The amplified RAPD PCR products were electrophoresed on a 1.0% agarose gel at 100 V for 30 min, and only ‘Asumi’-specific bands were excised. To select comparison markers, PCR products of approximately ≥700 bp, showing similar levels of amplification between ‘Asumi’ and ‘Asuki’, were excised. The excised PCR products were purified using the GgenAll^®^ Gel Purification Kit (GgenAll Biotechnology, Co., Seoul, Korea) and cloned using a pLUG-Prime^®^ TA-Cloning Vector Kit (iNtRON, Gyeonggi, Republic of Korea), as described by Jin et al. [52]. The nucleotide sequences of the cloned PCR products were determined by Solgent Co., Ltd. (Daejeon, Korea).

### 4.5. Development of Specific SCAR Markers and Comparison Markers for ‘Asumi’

To develop specific RAPD-SCAR markers and comparison markers for ‘Asumi’, the sequence information of genes was compared and analyzed using the BLAST program from NCBI. Based on the analysis results, seven sets of forward and reverse primers, ranging 20–31 bases in length, were designed for producing the RAPD-SCAR markers (Appendix A). For the comparison markers, five sets of forward primers, ranging 20–22 bases in length, and a 20-base reverse primer were designed (Appendix A).

### 4.6. Multiplex PCR Analysis

To facilitate the selection of SCAR markers and comparison markers, multiplex PCR was performed using six sets of primer mix combinations. These combinations included four sets of selected RAPD-SCAR markers and two sets of combination markers (control markers). The PCR reaction mixture consisted of AccuPower^®^ Multiplex PCR PreMix [Bioneer, Corp.; 250 μM dNTP, 1.5 mM MgCl_2_, 1.0 unit Taq DNA polymerase, 10 mM Tris-HCl (pH 9.0), and 40 mM KCl], 20–50 ng of genomic DNA, and 0.5 μM of each primer, adjusted to a final volume of 20 μL. The PCR reaction comprised 35 cycles, with denaturation at 94 °C for 5 min, followed by annealing at 94 °C for 30 s, 58 °C for 15 s, and 72 °C for 30 s, and a final extension at 72 °C for 5 min. The resulting PCR products were analyzed using the QiAxcel advanced system (Qiagen) electrophoresis device.

## 5. Conclusions

Herein, we successfully applied the developed multiplex PCR method to discriminate between ‘Asumi’ and ‘Asuki’ varieties, as well as other varieties with the genetic type of ponkan. This method allows for the rapid and accurate identification of varieties, even with limited resources. The implementation of this multiplex PCR method is expected to provide an efficient solution for cultivation farms facing challenges in cultivation management, harvest, and quality control due to unclear variety differentiation. By applying our established method for variety identification where ‘Asumi’ and ‘Asuki’ are intermixed, it will be possible to address the issues faced by cultivation farms, including challenges in cultivation and harvesting periods. Additionally, these selected markers make it possible to confirm the genetic origin of specific mandarin and Satsuma mandarin cultivars used as seed parents or pollen parents, and to easily determine whether the seedlings from crossbreeding are nucellar embryo seedlings derived from the seed parent or zygotic seedlings.

## Figures and Tables

**Figure 1 plants-12-03988-f001:**
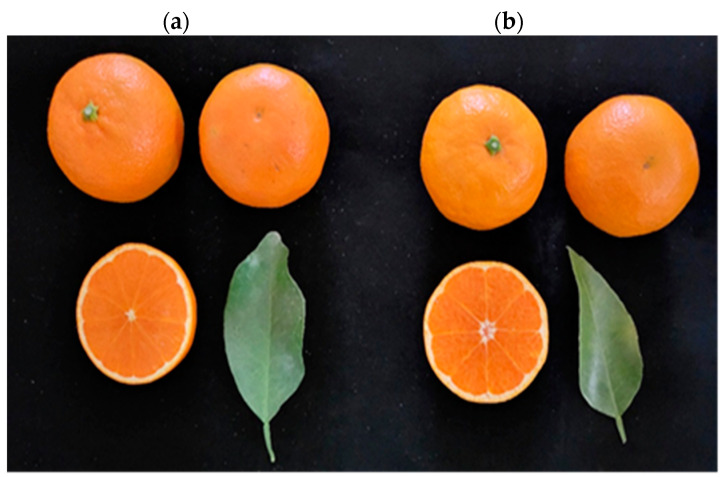
Morphology of (**a**) ‘Asumi’ and (**b**) ‘Asuki’ fruits.

**Figure 2 plants-12-03988-f002:**
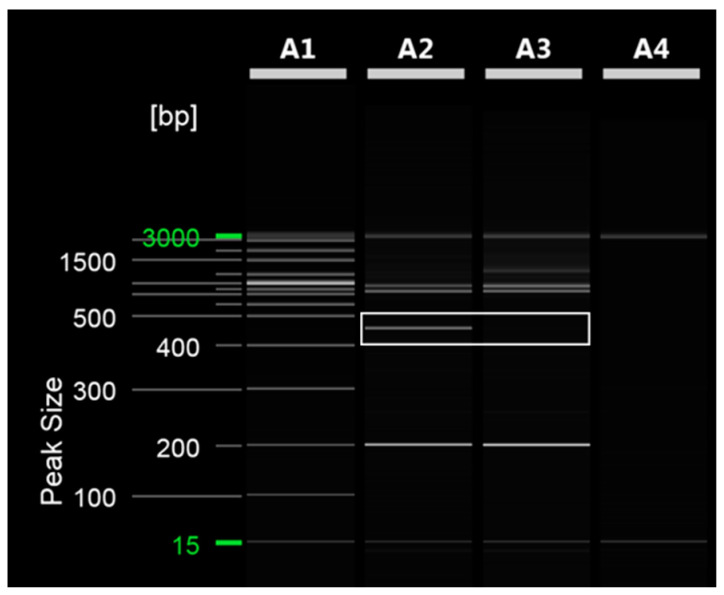
RAPD PCR analysis of ‘Asumi’ and ‘Asuki’ cultivars. Polyacrylamide gel electrophoresis was performed using the QiAxcel advanced system. The description of lanes is as follows: A1: molecular markers (i.e., 20 bp and 100 bp DNA Ladders, Qiagen); A2: *Citrus* hybrid ‘Asumi’ cultivar; A3: *Citrus* hybrid ‘Asuki’ cultivar; A4: negative control.

**Figure 3 plants-12-03988-f003:**
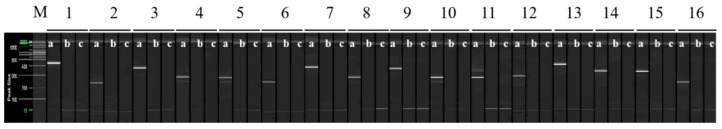
The RAPD-SCAR markers used to select ‘Asumi’. Polyacrylamide gel electrophoresis was performed using the QiAxcel advanced system. The description of lanes is as follows: M: molecular markers (i.e., 20 bp and 100 bp DNA Ladders, Qiagen); 1: RAPD-SCAR UBC 218F and 218R primer set combination; 2: RAPD-SCAR UBC F1 and R3 primer set combination; 3: RAPD-SCAR UBC F1 and R1-1 primer set combination; 4: RAPD-SCAR UBC F1 and R2-1 primer set combination, 5: RAPD-SCAR UBC F1 and R3-1 primer set combination, 6: RAPD-SCAR UBC F2 and R3 primer set combination, 7: RAPD-SCAR UBC F2 and R1-1 primer set combination, 8: RAPD-SCAR UBC F2 and R3-1 primer set combination 9: RAPD-SCAR UBC F3 and R1-1 primer set combination, 10: RAPD-SCAR UBC F3 and R2-1 primer set combination, 11: RAPD-SCAR UBC F3 and R3-1 primer set combination, 12: RAPD-SCAR UBC F1-1 and R3 primer set combination, 13: RAPD-SCAR UBC F1-1 and R1-1 primer set combination, 14: RAPD-SCAR UBC F1-1 and R2-1 primer set combination, 15: RAPD-SCAR UBC F1-1 and R3-1 primer set combination, 16: RAPD-SCAR UBC F2-1 and R1 primer set combination. The description of sublanes is as follows: a: *Citrus* hybrid ‘Asumi’; b: *Citrus* hybrid ‘Asuki’; c: negative control.

**Figure 4 plants-12-03988-f004:**
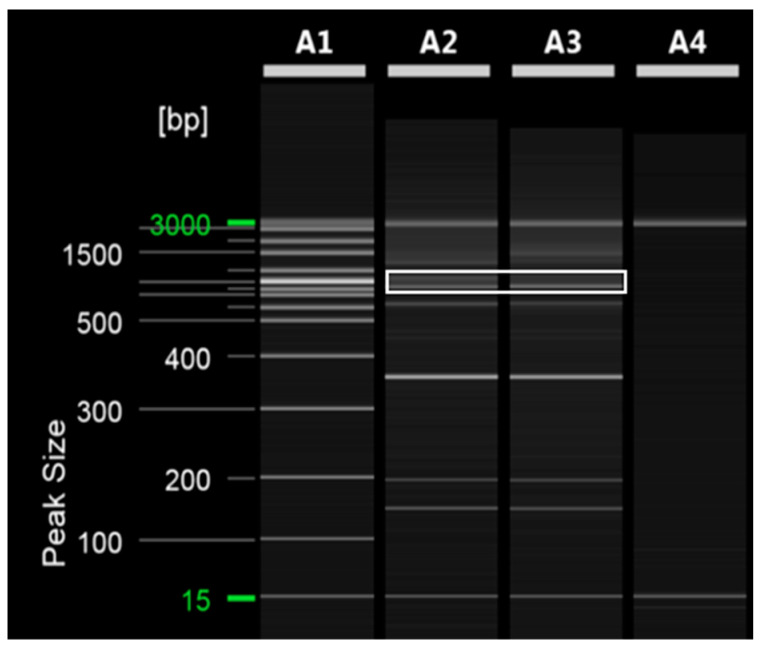
SRAP PCR analysis of amplification in both ‘Asumi’ and ‘Asuki’ cultivars. Polyacrylamide gel electrophoresis was performed using the QiAxcel advanced system. The description of lanes is as follows: A1: molecular markers (i.e., 20 bp and 100 bp DNA Ladders, Qiagen); A2: *Citrus* hybrid ‘Asumi’; A3: *Citrus* hybrid ‘Asuki’; A4: negative control.

**Figure 5 plants-12-03988-f005:**
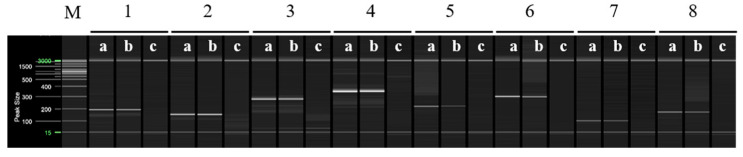
SRAP-SCAR PCR results for control marker selection. Polyacrylamide gel electrophoresis was performed using the QiAxcel advanced system. The description of lanes is as follows: M: molecular markers (i.e., 20 bp and 100 bp DNA Ladders, Qiagen); 1: SRAP-SCAR F2 and R2 primer set combination; 2: SRAP-SCAR F2 and R3 primer set combination; 3: SRAP-SCAR F2 and R4 primer set combination; 4: SRAP-SCAR F2 and R5 primer set combination, 5: SRAP-SCAR F3 and R2 primer set combination, 6: SRAP-SCAR F3 and R4 primer set combination, 7: SRAP-SCAR F5 and R4 primer set combination, 8: SRAP-SCAR F5 and R5 primer set combination. The description of sublanes is as follows: a: *Citrus* hybrid ‘Asumi’; b: *Citrus* hybrid ‘Asuki’; c: negative control.

**Figure 6 plants-12-03988-f006:**
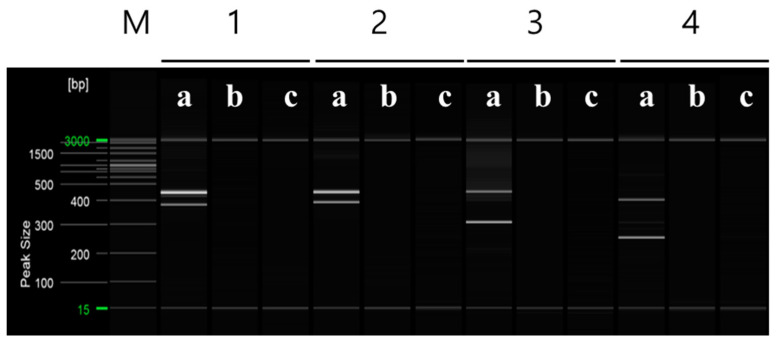
Multiplex PCR products amplified from ‘Asumi’ and ‘Asuki’ cultivars using two pairs of RAPD-SCAR primers. Polyacrylamide gel electrophoresis was performed using the QiAxcel advanced system. The description of lanes is as follows: M: molecular markers (20 bp and 100 bp DNA Ladders, Qiagen); 1: No. 1 and No. 3 primer set combination; 2: No. 1 and No. 7 primer set combination; 3: No. 12 and No. 13 primer set combination; 4: No. 13 and No. 15 primer set combination. The description of sublanes is as follows: a: *Citrus* hybrid ‘Asumi’; b: *Citrus* hybrid ‘Asuki’; c: negative control.

**Figure 7 plants-12-03988-f007:**
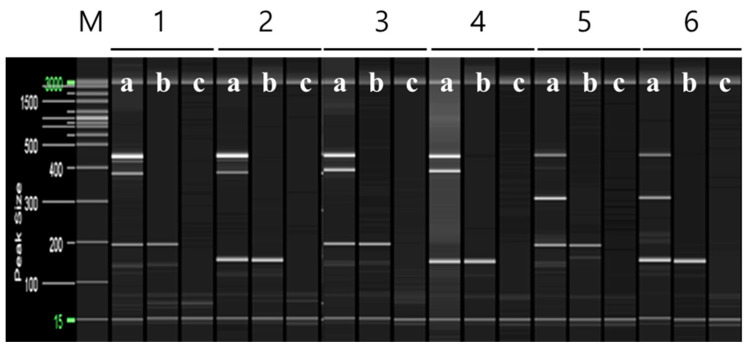
Multiplex PCR products amplified from ‘Asumi’ and ‘Asuki’ cultivars using two pairs of RAPD-SCAR primers and one set of control primers. Polyacrylamide gel electrophoresis was performed using the QiAxcel advanced system. The description of lanes is as follows: M: molecular markers (20 bp and 100 bp DNA Ladders, Qiagen); 1: No. 1, No. 3, and SRAP F2/R2 primer set combination; 2: No. 1, No. 3, and SRAP F2/R3 primer set combination; 3: No. 1, No. 7, and SRAP F2/R2 primer set combination; 4: No. 1, No. 7, and SRAP F2/R3 primer set combination; 5: No. 12, No. 13, and SRAP F2/R2 primer set combination; 6: No. 12, No. 13, and SRAP F2/R3 primer set combination. The description of sublanes is as follows: a: *Citrus* hybrid ‘Asumi’; b: *Citrus* hybrid ‘Asuki’; c: negative control.

**Figure 8 plants-12-03988-f008:**
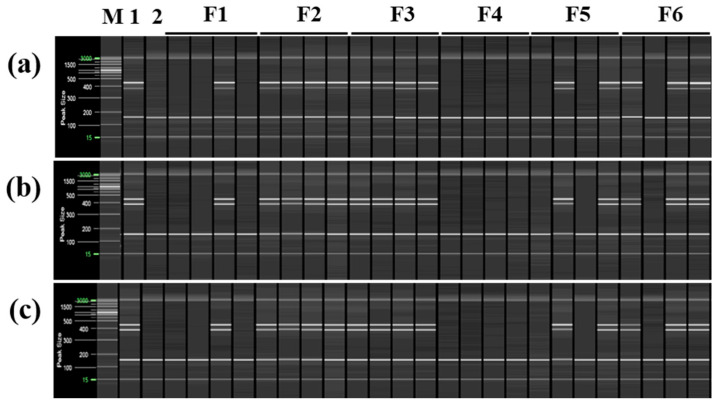
Identification of ‘Asumi’ among citrus cultivars from different farmhouses using two pairs of RAPD-SCAR primers and one set of control primers. Polyacrylamide gel electrophoresis was performed using the QiAxcel advanced system. The description of rows is as follows: (**a**): SRAP F2/R3 primer set combination; (**b**): No. 1, No. 7, and SRAP F2/R3 primer set combination; (**c**): No. 12, No. 13, and SRAP F2/R3 primer set combination. The description of lanes is as follows: M: molecular markers (i.e., 20 bp and 100 bp DNA Ladders, Qiagen); 1: *Citrus* hybrid ‘Asumi’; 2: *Citrus* hybrid ‘Asuki’; F1–F6: Farmhouses 1–6.

**Figure 9 plants-12-03988-f009:**
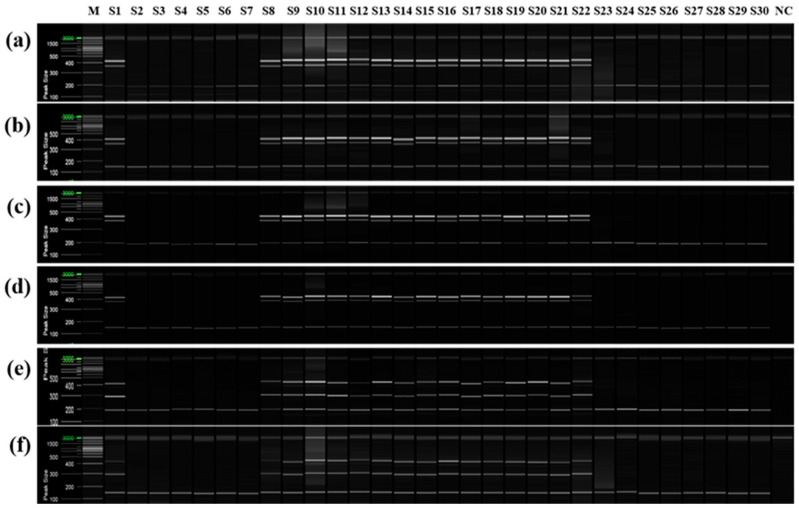
Multiplex PCR products amplified from mandarin cultivars and hybrid cultivars using two pairs of RAPD-SCAR primers and one set of control primers. Polyacrylamide gel electrophoresis was performed using the QiAxcel advanced system. The description of rows is as follows: (**a**): No. 1, No. 3, and SRAP F2/R2 primer set combination; (**b**): No. 1, No. 3, and SRAP F2/R3 primer set combination; (**c**): No. 1, No. 7, and SRAP F2/R2 primer set combination; (**d**): No. 1, No. 7, and SRAP F2/R3 primer set combination; (**e**): No. 12 No. 13, and SRAP F2/R2 primer set combination; (**f**): No. 12, No. 13, and SRAP F2/R3 primer set combination. The description of lanes is as follows: M: molecular markers (20 bp and 100 bp DNA Ladders, Qiagen); S1: ‘Asumi’; S2: ‘Asuki’; S3: ‘Okitsu wase’; S4: ‘Miyagawa wase’; S5: ‘Nichinan 1gou’; S6: ‘Haryejosaeng’; S7: ‘Kiyomi’; S8: ‘Wilking’; S9: ‘Nova’; S10: ‘Lee’; S11: ‘Encore’; S12: ‘Ootaponkan’; S13: ‘Hayaka Ponkan’; S14: ‘Murcott’; S15: ‘Harumi’; S16: ‘Shiranuhi’; S17: ‘Kanpei’; S18: ‘Setoka’; S19: ‘Haruka’; S20: ‘Winter Prince’; S21: ‘Tamnaneunbong’; S22: ‘Yellow ball’; S23: ‘Minihyang’; S24: ‘Tsunokaori’; S25: ‘Sinyegam’; S26: ‘Mihaya’; S27: ‘Miraehyang’; S28: ‘Eime Kashi 28 gou’; S29: ‘Pungkwang navel’; S30: ‘Washingon navel’; NC: negative control.

**Figure 10 plants-12-03988-f010:**
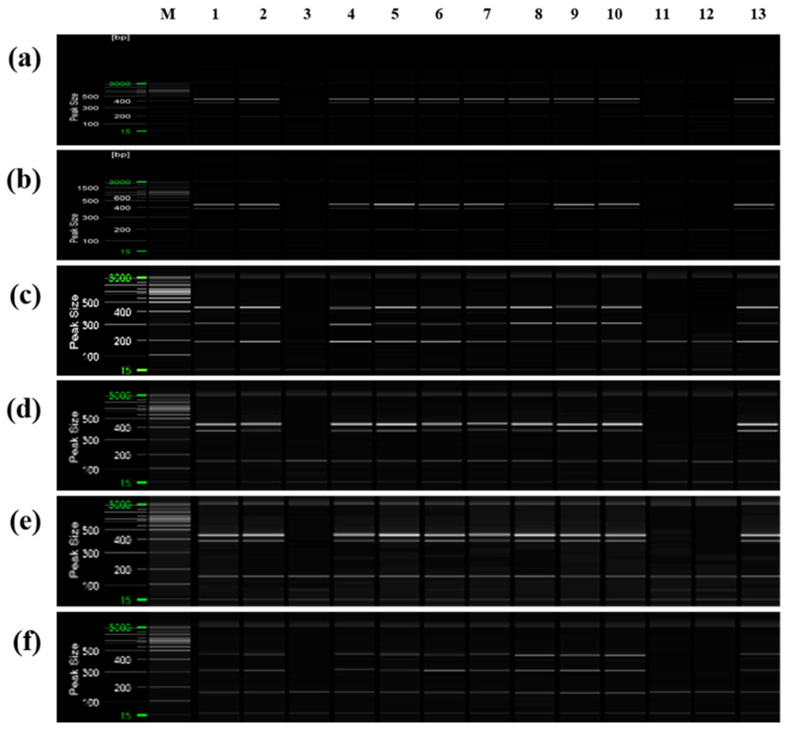
Multiplex PCR products amplified from mandarin cultivars using two pairs of RAPD-SCAR primers and one set of control primers. Polyacrylamide gel electrophoresis was performed using the QiAxcel advanced system. The description of rows is as follows: (**a**): No. 1, No. 3, and SRAP F2/R2 primer set combination; (**b**): No. 1, No. 3, and SRAP F2/R3 primer set combination; (**c**): No. 1, No. 7, and SRAP F2/R2 primer set combination; (**d**): No. 1, No. 7, and SRAP F2/R3 primer set combination; (**e**): No. 12, No. 13, and SRAP F2/R2 primer set combination; (**f**): No. 12, No. 13, and SRAP F2/R3 primer set combination. The description of lanes is as follows: M: molecular markers (20 bp and 100 bp DNA Ladders, Qiagen); 1: ’Dancy’; 2: ‘Natsumi’; 3: ‘Byungkyul’; 4: ‘Jinkyul’; 5: ‘Binkyul’; 6: ‘Southern Yellow’; 7: ‘King’; 8: ‘Kinnow’; 9: ‘Dongjeonkyul’; 10: ‘Fortune’; 11: ‘Page’; 12: ‘Kinokuni’; and 13: ‘Clementine’.

**Table 1 plants-12-03988-t001:** Comparative analysis of sugar content and acidity level between ‘Asumi’ and ‘Asuki’.

Cultivar	Sugar Content (Brix)	Acidity Level (%)
‘Asumi’	15.58 ± 0.65	1.10 ± 0.11
‘Asuki’	15.14 ± 0.41	1.92 ± 0.25

## Data Availability

The datasets generated and/or analyzed during the current study are available from the corresponding author upon reasonable request.

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
