# Peer review of "Development of Pollen Parent Cultivar-Specific SCAR Markers and a Multiplex SCAR-PCR System for Discrimination between Pollen Parent and Seed Parent in Citrus"

_plants, 2023, doi:10.3390/plants12233988_

Round 1

Reviewer 1 Report

Comments and Suggestions for Authors

The MS discusses the challenge of distinguishing between high-quality mandarin cultivars, "Asumi" and "Asuki," which were developed through crossbreeding in Korea. Due to genetic similarity resulting from crossbreeding between the same cultivars, it is challenging to differentiate them morphologically at the seedling stage. This difficulty poses challenges for cultivation and harvesting on farms. To address this issue, the authors developed a method using SCAR markers for rapid and accurate differentiation between the two cultivars. They selected specific primer sets from random amplified polymorphic DNA–SCAR combinations and sequence-related amplified polymorphism contrast markers. The multiplex PCR system using these markers was able to identify 16 mandarin cultivars, including "Asumi" and "Asuki," among 30 cultivars. The use of these SCAR markers is expected to enhance citrus cultivation by accurately identifying mixed cultivars and facilitating proper harvest timing for citrus distribution. Additionally, the markers can help identify the genetic traits of hybrid varieties at the seedling stage.

Some comments:

  1. The abstract provides a clear overview of the study, outlining the issue, the developed method using SCAR markers, and the expected benefits. However, the structure could be improved by organizing information more systematically.
  2. The text uses technical terms such as SCAR markers, primer sets, and multiplex PCR system without providing explanations. This may make it challenging for readers unfamiliar with molecular biology and genetics to fully grasp the content.
  3. The abstract and other sections mention the difficulty in distinguishing between cultivars but does not elaborate on the specific challenges faced by farmers or the consequences of misidentification. Providing more context on the real-world implications would enhance the relevance of the information.
  4. While the text explains the method developed using SCAR markers, it lacks information on in-depth analysis of the experimental design, statistical methods used, or potential limitations of the study. Including these aspects would strengthen the academic quality.

5.      The MS does ma include relevant scientific studies to embolden the discussion and introduction of the related research using existing literature, which is crucial in academic writing to establish context and credibility. For example, see See below few examples from literature on molecular marker-based analysis (using RAPD, RFLP, AFLP, SSR, ISSR, ITS, etc): PMID: 37510337; PMID: 35562398;  Multiplex molecular marker-assisted analysis of significant pathogens of cotton (Gossypium sp.), 2022; Biocatalysis and Agricultural Biotechnology https://doi.org/10.1016/j.bcab.2022.102557 (Cotton);  Microsatellite and RAPD analysis of grape (Vitis spp.) accessions and identification of duplicates/misnomers in germplasm collection, Upadhyay et al., 2010 Indian J Hortic Volume 67 Pages 8-15; Microsatellite analysis to differentiate clones of Thompson seedless grapevine, Upadhyay et al., 2010, Ind Journal of Horticulture, Volume 67 Issue 2 Pages 260-263.

  1. The summary briefly touches on the expected benefits of using SCAR markers but does not elaborate on the broader implications of this research in the field of citrus cultivation or genetic studies.
  2. Table 2 and 3; Figure 3 and 6 could be moved to supplementary information.

In summary, while the MS presents valuable information on addressing the challenges in distinguishing between mandarin cultivars, improvements in clarity, technical language explanation, relevance, depth of analysis, citation of references, and a more comprehensive conclusion would enhance its academic quality.

Comments on the Quality of English Language

Rewriting some texts to reduce technical language will hep the readers.

Author Response

Reviewer 1.

  1. The abstract provides a clear overview of the study, outlining the issue, the developed method using SCAR markers, and the expected benefits. However, the structure could be improved by organizing information more systematically.

> Improvements were made through content revisions, and the revised content was marked in red and submitted.

  1. The text uses technical terms such as SCAR markers, primer sets, and multiplex PCR system without providing explanations. This may make it challenging for readers unfamiliar with molecular biology and genetics to fully grasp the content.

> The technical terminology was revised to make it easier to understand, and the revised content was marked in red and submitted.

  1. The abstract and other sections mention the difficulty in distinguishing between cultivars but does not elaborate on the specific challenges faced by farmers or the consequences of misidentification. Providing more context on the real-world implications would enhance the relevance of the information.

> The abstract was revised, supplemented, and submitted with the revised content marked in red. 4. While the text explains the method developed using SCAR markers, it lacks information on in-depth analysis of the experimental design, statistical methods used, or potential limitations of the study. Including these aspects would strengthen the academic quality.

  1. The MS does ma include relevant scientific studies to embolden the discussion and introduction of the related research using existing literature, which is crucial in academic writing to establish context and credibility. For example, see See below few examples from literature on molecular marker-based analysis (using RAPD, RFLP, AFLP, SSR, ISSR, ITS, etc): PMID: 37510337; PMID: 35562398; Multiplex molecular marker-assisted analysis of significant pathogens of cotton (Gossypium sp.), 2022; Biocatalysis and Agricultural Biotechnology https://doi.org/10.1016/j.bcab.2022.102557 (Cotton); Microsatellite and RAPD analysis of grape (Vitis spp.) accessions and identification of duplicates/misnomers in germplasm collection, Upadhyay et al., 2010 Indian J Hortic Volume 67 Pages 8-15; Microsatellite analysis to differentiate clones of Thompson seedless grapevine, Upadhyay et al., 2010, Ind Journal of Horticulture, Volume 67 Issue 2 Pages 260-263.

> References have been modified by additions and replacements, and the revised content is marked in red.

  1. The summary briefly touches on the expected benefits of using SCAR markers but does not elaborate on the broader implications of this research in the field of citrus cultivation or genetic studies.

 > Improvements were made through content revisions, and the revised content was marked in red and submitted.

  1. Table 2 and 3; Figure 3 and 6 could be moved to supplementary information.

> Photos(3 and 6) and tables (2 and 3) were moved to supplementary material and the contents were modified.

  1. In summary, while the MS presents valuable information on addressing the challenges in distinguishing between mandarin cultivars, improvements in clarity, technical language explanation, relevance, depth of analysis, citation of references, and a more comprehensive conclusion would enhance its academic quality.

>Improvements were made through content revisions, and the revised content was marked in red and submitted.

Reviewer 2 Report

Comments and Suggestions for Authors

The present document develops a method to discriminate seed coming from cross pollination and polyembrionic seeds coming from the parent.

The title, is a little long and confusing, and do not include Citrus.

The abstract is clear.

The introduction overview clearly the topic and introduces clearly the problem, only references 2-3 when first citated are not appropriated to describe the economic market of citrus.

The results are clearly presented.

The discussion is clear and complete.

In the material and method section: the analysis of sugar and acidity performed is missing, even is not the main goal and are classic analysis you should mention it.

The conclusion correspond to the results obtained.

Minor form corrections can be found in the attached document

Author Response

Review-2

The present document develops a method to discriminate seed coming from cross pollination and polyembrionic seeds coming from the parent.

  1. The title, is a little long and confusing, and do not include Citrus.

>Improvements were made by editing the title, and the revised content was marked in red and submitted.

  1. The introduction overview clearly the topic and introduces clearly the problem, only references 2-3 when first citated are not appropriated to describe the economic market of citrus.

> It was revised with references appropriate to the content, and the revised content was marked in red and submitted.

  1. In the material and method section: the analysis of sugar and acidity performed is missing, even is not the main goal and are classic analysis you should mention it.

>Contents related to fruit sugar acid analysis were attached and revised in the materials and methods, and the revised content was marked in red and submitted.

Reviewer 3 Report

Comments and Suggestions for Authors

11 Cultivars should be in single quotes 

11 Crossbreeding between the same cultivar makes no sense

15 Define SCARs

22 This sentence is not correct

29 omit family  aceae is understood as family

46 ripens where?

52 At this point I believe that the authors require expert language assistance.  Otherwise,

Much of the meaning is obscured or lost.

69  References should be paired with the technique.

Comments on the Quality of English Language

Needs to have an English language expert review the manuscript for clarity.

Author Response

Reviewer 3

  1. 11 Cultivars should be in single quotes

> All cultivar names were modified with single quotation marks.

  1. 11 Crossbreeding between the same cultivar makes no sense

>Improvements were made through content revisions, and the revised content was marked in red and submitted.

  1. 15 Define SCARs

>It has been revised to make its meaning clear, and the revised content has been marked in red and submitted.

  1. 22 This sentence is not correct

> It has been revised to make its meaning clear, and the revised content has been marked in red and submitted.

  1. 46 ripens where?

>Content related to fruit harvest has been modified.

  1. 52 At this point I believe that the authors require expert language assistance. Otherwise,

Much of the meaning is obscured or lost.

>The overall contents have been revised.

  1. 69 References should be paired with the technique.

>References have been revised and overall improvements have been made.
